# Image-Less THA Cup Navigation in Clinical Routine Setup: Individual Adjustments, Accuracy, Precision, and Robustness

**DOI:** 10.3390/medicina58060832

**Published:** 2022-06-20

**Authors:** Corinne A. Zurmühle, Benjamin Zickmantel, Matthias Christen, Bernhard Christen, Guoyan Zheng, Joseph M. Schwab, Moritz Tannast, Simon D. Steppacher

**Affiliations:** 1Department of Orthopaedic Surgery and Traumatology, HFR Hôpital Cantonal, University of Fribourg, 1700 Fribourg, Switzerland; 2Department of Orthopaedic Surgery and Traumatology, Kantonsspital Aarau AG, 5001 Aarau, Switzerland; benjamin.zickmantel@ksa.ch; 3Department of Orthopaedic Surgery, Inselspital, Bern University Hospital, University of Bern, 3010 Bern, Switzerland; matthias.christen@outlook.com (M.C.); simondamian.steppacher@insel.ch (S.D.S.); 4Articon Spezialpraxis für Gelenkchirurgie, Salem-Spital, 3013 Bern, Switzerland; b.christen@articon.ch; 5Institute of Medical Robotics, School of Biomedical Engineering, Shanghai Jiao Tong University, Shanghai 200240, China; guoyan.zheng@sjtu.edu.cn; 6Department of Orthopaedic Surgery, Medical College of Wisconsin, Milwaukee, WI 53226, USA; jmschwab@mcw.edu

**Keywords:** cup positioning, image–free navigated THA, 2D/3D matching, accuracy, precision, intraoperative adjustments

## Abstract

*Background and Objectives*: Even after the ‘death’ of Lewinnek’s safe zone, the orientation of the prosthetic cup in total hip arthroplasty is crucial for success. Accurate cup placement can be achieved with surgical navigation systems. The literature lacks study cohorts with large numbers of hips because postoperative computer tomography is required for the reproducible evaluation of the acetabular component position. To overcome this limitation, we used a validated software program, HipMatch, to accurately assess the cup orientation based on an anterior-posterior pelvic X-ray. The aim of this study were to (1) determine the intraoperative ‘individual adjustment’ of the cup positioning compared to the widely suggested target values of 40° of inclination and 15° of anteversion, and evaluate the (2) ‘accuracy’, (3) ‘precision’, and (4) robustness, regarding systematic errors, of an image-free navigation system in routine clinical use. *Material and Methods:* We performed a retrospective, accuracy study in a single surgeon case series of 367 navigated primary total hip arthroplasties (PiGalileo^TM^, Smith+Nephew) through an anterolateral approach performed between January 2011 and August 2018. The individual adjustments were defined as the differences between the target cup orientation (40° of inclination, 15° of anteversion) and the intraoperative registration with the navigation software. The accuracy was the difference between the intraoperative captured cup orientation and the actual postoperative cup orientation determined by HipMatch. The precision was analyzed by the standard deviation of the difference between the intraoperative registered and the actual cup orientation. The outliers were detected using the Tukey method. *Results:* Compared to the target value (40° inclination, 15° anteversion), the individual adjustments showed that the cups are impacted in higher inclination (mean 3.2° ± 1.6°, range, (−2)–18°) and higher anteversion (mean 5.0° ± 7.0°, range, (−15)–23°) (*p* < 0.001). The accuracy of the navigated cup placement was −1.7° ± 3.0°, ((−15)–11°) for inclination, and −4.9° ± 6.2° ((−28)–18°) for anteversion (*p* < 0.001). Precision of the system was higher for inclination (standard deviation SD 3.0°) compared to anteversion (SD 6.2°) (*p* < 0.001). We found no difference in the prevalence of outliers for inclination (1.9% (7 out of 367)) compared to anteversion (1.63% (6 out of 367), *p* = 0.78). The Bland-Altman analysis showed that the differences between the intraoperatively captured final position and the postoperatively determined actual position were spread evenly and randomly for inclination and anteversion. *Conclusion:* The evaluation of an image-less navigation system in this large study cohort provides accurate and reliable intraoperative feedback. The accuracy and the precision were inferior compared to CT-based navigation systems particularly regarding the anteversion. However the assessed values are certainly within a clinically acceptable range. This use of image-less navigation offers an additional tool to address challenging hip prothesis in the context of the hip–spine relationship to achieve adequate placement of the acetabular components with a minimum of outliers.

## 1. Introduction

Even after the ‘death’ of Lewinnek’s safe zone [1], the orientation of the acetabular component in total hip arthroplasty remains a key factor for success. Currently, concepts such as the hip-spine relationship, postoperative psoas impingement problems, and the combined version fueled additional interest into placing acetabular components. The accurate placement and orientation of the cup can be achieved by surgical navigation systems. The reliable and precise evaluation of these systems requires postoperative computed tomography, which remains the gold standard. However, it substantially limits the number of hips in validation studies because of the associated costs and unnecessary radiation exposure for patients. Larger validation studies are based on postoperative radiographs only, which may be subject to significant errors [2]. In fact, a study on the navigated cup orientation in a large cohort of THAs representing daily clinical practice and covering several years is lacking.

To overcome this limitation, the software program, HipMatch [3,4], was developed and validated. HipMatch allows the accurate calculation of cup orientation relative to the anterior pelvic plane based on uncalibrated postoperative anteroposterior pelvis X-rays. This enables us to analyze a larger series of navigated acetabular components in daily practice since these images are acquired routinely, whereas postoperative CT scans are not.

Using HipMatch software, the main aim of this study was to (1) determine the intraoperative ‘individual adjustment’ of the cup positioning compared to the widely suggested target values of 40° of inclination and 15° of anteversion. In addition, we evaluated the (2) ‘accuracy’, (3) ‘precision’, and (4) robustness regarding systematic errors of an image-free navigation system in routine clinical use (Figure 1). The actual postoperative cup orientation determined using HipMatch was used as the source of the truth.

## 2. Material and Methods

We performed a retrospective, accuracy study in a single surgeon (CB) case series of 649 consecutive navigated primary hip total arthroplasties performed between January 2011 and August 2018 at a single institution. We excluded a total of 282 hips due to missing navigation data (105 hips), inadequate postoperative radiographs (149 hips) or problems with 3D modeling (28 hips, e.g., in severe pelvic deformities) preventing postoperative digital evaluation using HipMatch (Figure 2). Finally, 367 hips (340 patients) were included for further analysis (Table 1).

Indications for surgery were primary hip osteoarthritis in 323 hips and secondary osteoarthritis in 44 hips (15 developmental dysplasia of the hip, 21 avascular necrosis and eight sequelae of femoral neck fractures). Fifteen hips had previous surgery: three femoral osteotomies, five reductions and internal fixations of proximal femur fractures, four surgical hip dislocations for femoroacetabular impingement syndrome, two hip arthroscopies and one monopolar hip arthroplasty (Table 1). During the follow-up patients showed the following complications: 0.3 % luxation (1/367 hips), 0.3% loosening of the stem (1/367 hips), 2% infection (7/367 hips), and 0.5% seroma (2/367 hips).

All total hip arthroplasties were performed through an anterolateral approach (Watson-Jones) with the patient positioned in supine position. The acetabular component used was a press-fit cup (EP-FIT PLUS^TM^, Smith & Nephew). Image-free navigation (PiGalileo^TM^, Smith+Nephew) was applied with passive digital reference based on references of the pelvic wing and the distal femur. The anterior pelvic plane (APP) constructed by the two anterior superior iliac spines and the symphysis (defined as the midpoint between the pubic tubercles) was manually digitized using the navigation system intraoperatively and served as the anatomical reference coordinate system. The inclination and anteversion calculated from the navigation system was based on the radiological definition [5]. Generally, the target cup orientation was set as 15° of anteversion and 40° of inclination [6,7]. This initial cup orientation was then adapted according to the individual morphology of the acetabular rim, such as to avoid psoas tendon irritation. After final impaction of the cup, the intraoperative cup orientation, displayed and calculated from the navigation system (the ‘final cup orientation’), was captured. Postoperatively, an anteroposterior pelvic radiograph centered on the symphysis was obtained in all cases. The ‘actual cup orientation’ was calculated based on this radiograph using HipMatch [3]. Using statistical shape modeling, the software creates a virtual 3D model of the pelvis with spatial information of the APP and, thus, allows an anatomically-based calculation of anteversion and inclination with respect to this plane. The reported accuracy of this system was of 0.4 ± 1.8 (−2.6° to 3.3°) for inclination and 0.6 ± 1.5° (−2.0 to 3.9°) for anteversion, with excellent reproducibility and reliability [3].

## 3. Statistics

For question 1, we defined the ‘individual adjustment’ as the difference between the target cup orientation (40° of inclination and 15° of anteversion) and the final cup orientation displayed by the navigation system intraoperatively after the adjustment of the cup and the final impaction (Table 2). After confirming the normal distribution with the Kolmogorov-Smirnov test, we compared the differences using an unpaired Student’s *t*-test.

For question 2, we defined the ‘accuracy’ as the difference between the intraoperative captured final cup orientation and the postoperative actual cup orientation, which were determined using HipMatch as the gold standard. For question 1 and 2, values were calculated as the mean difference using the standard deviation and range.

For question 3, we defined the ‘precision’ as how close the intraoperative registration was to the actual cup orientation. We used the standard deviation of the difference between the intraoperatively registered and the actual cup orientation for evaluation. We used the F-test to compare the potential differences between the standard deviations. In addition, the percentage of outliers (identified with the Tukey method [8]) was compared using Fisher’s exact test. The statistical analysis was performed on MedCalc for Windows, version 20.015 (MedCalc Software Ltd., Ostend, Belgium).

For question 4, a Bland–Altman analysis was performed to evaluate the robustness regarding systematic errors [9].

## 4. Results

### 4.1. Individual Adjustment

After individual adjustment to the patient’s anatomy, the cups were generally impacted in higher inclination (mean 3.2° ± 1.6°, range, (−2)–18°) compared to the target inclination of 40°. Similarly, the cups were impacted in higher anteversion (mean 5.0° ± 7.0°, range, (−15)–23°) compared to the target anteversion of 15°. This difference was significantly higher for the anteversion (*p* < 0.001, Figure 3).

### 4.2. Accuracy

The accuracy of the navigated cup placement was −1.7° ± 3.0°, ° for the inclination, and the range was, (−15)–11°. The accuracy was −4.9° ± 6.2°, for the anteversion, and the range was (−28)–18°. This difference was significantly higher for the anteversion (*p* < 0.001, Figure 4).

### 4.3. Precision

The precision of the navigation system was significantly higher for the inclination (standard deviation 3.0°) compared to the anteversion (6.2°, *p* < 0.001, Figure 4)

We found no difference in the prevalence of outliers for inclination (1.9% [7 out of 367] compared to anteversion (1.63% [6 out of 367], *p* = 0.78, Figure 4).

### 4.4. Robustness

The Bland-Altman analysis showed that the differences between the intraoperatively captured final position and the postoperatively determined actual position were spread evenly and randomly for the inclination (Figure 5A) and the anteversion (Figure 5B).

## 5. Discussion and Conclusions

The evaluation of the accuracy, precision, and robustness of commercially available navigation systems in large series of patients is limited since it has historically required a postoperative CT scan. The HipMatch software allows a fast, anatomically-based evaluation of the actual cup position based on a standard postoperative X-ray. We applied this technique in one of the largest study cohorts to validate the accuracy of navigation systems. Based on a default target value of 40° of inclination and 15° of anteversion, we observed that–after individual intraoperative adjustment–the cups were typically impacted at a higher inclination and, particularly, a higher anteversion. The accuracy of the final displayed intraoperative cup orientation was significantly higher for the inclination than for the anteversion, as was the precision. There were no systematic errors or differences in the prevalence of outliers.

This study has some limitations. It comprises data from a single–surgeon series only. In general, this allows a higher familiarity with the navigation system and eliminates user-dependent technical problems. However, the final cup orientation depends on the individual judgement of a single (high level THA) surgeon. The interpretation of our first question is therefore limited to the individual preference of the surgeon.

Referring to question 1, we observed a substantial individual adjustment of the cup orientation intraoperatively. The cups were generally impacted at a 5° greater anteversion and a somewhat greater inclination compared to the default cup orientation (40° of inclination and 15° of anteversion). The essential aim was to match the prosthetic cup orientation to the native orientation of the acetabulum. This ensures the optimal pressfit of the cup and avoids secondary impingement of the iliopsoas tendon. The native anteversion is approximately 20° [10], which may explain the 5° difference in the anteversion during the intraoperative adjustment in our study. The differences in the inclination were clinically irrelevant which can be explained by the fact that the native Sharp angle in a normal hip is approximately 40°, corresponding well to the target inclination [11]. In addition, in some of the cases with torsional problems of the femur, the anteversion was adjusted to obtain a normal McKibbin-index (combined femoral and acetabular version). This phenomenon corresponds well to previous studies comparing the evaluation of free-hand cup orientation [12,13].

Referring to question 2, we found a significantly higher accuracy of the navigation system for the inclination compared to the anteversion. This is consistent with other accuracy studies (Table 3 [13,14,15,16,17,18,19]). The reasons for this are twofold. First, the inclination is mainly defined by capturing the anterior superior iliac spines, which are further apart in space and therefore provide a more robust reference for the anterior pelvic plane. By contrast, the definition of the pubic symphysis has been shown to be subject to potential error, especially in the anteroposterior direction, which influences the anteversion angle to a substantial amount. Second, using Murray’s [5] nomograms on interrelationship between the anteversion and inclination, it is a mathematical effect that a change of the same amount for inclination of about 40° leads to a larger change of anteversion, especially in anteversion ranging around 15°. This may explain the differences observed.

Referring to question 3, the precision of the navigation system was higher for the inclination than for the anteversion. Again, this is a fact that corresponds to the findings in the literature (Table 3). Analogous to the accuracy, the cup anteversion is highly reliant on the definition of the mid pubis point. While the anterior superior iliac spine can be digitized easily, the percutaneous definition of the mid pubis point implies a certain inaccuracy especially in image-less navigation systems [13,20] due to the prepubic soft tissue (particularly in more obese patients). This might explain the decreased precision reported in the free-hand navigation systems compared to CT-based navigation systems [21]. However, even with a large group of patients in routine clinical practice, we were able to show that the accuracy was quite comparable to smaller prospective studies in an optimized study design.

Referring to question 4, we evaluated the prevalence of outliers. In contrast to many previous studies, we did not define outliers using Lewinnek’s safe zone as this definition is no longer considered a relevant target [1]. Instead, our approach was based on any outlier identified with the Tukey method [8], which checks for multiple outliers on either side. This allowed a more objective and less arbitrary identification of outliers. We are certain that in clinical practice it is more important to reduce the ‘real’ (statistically defined) outliers than to refer to the “safe zone”, which has been criticized. Compared to the literature, our approach seems to underline the change in the paradigm from the ‘safe zone’ to the ‘reduction of outliers’.

In summary, the image-less navigation system provides accurate and reliable intraoperative feedback, which is often adjusted individually during surgery to individual anatomy e.g for reducing psoas impingement without an elevated number of statistical outliers. The accuracy and precision of the image-less navigation system are inferior compared to CT-based navigation systems particularly for the anteversion. However, the values we obtained are certainly within a clinically acceptable range. This use of image-less navigation offers the orthopedic surgeon an additional tool to address challenging hip prothesis in the context of the hip-spine-relationship or combined version to achieve the adequate placement of the acetabular components with a minimum of outliers. This justifies its use in a routine clinical practice. In the future long-term studies, a correlation of the individual cup positioning and the outliers with wear or prosthetic loosing of the acetabular cup may deepen the knowledge in this field.

## Figures and Tables

**Figure 1 medicina-58-00832-f001:**
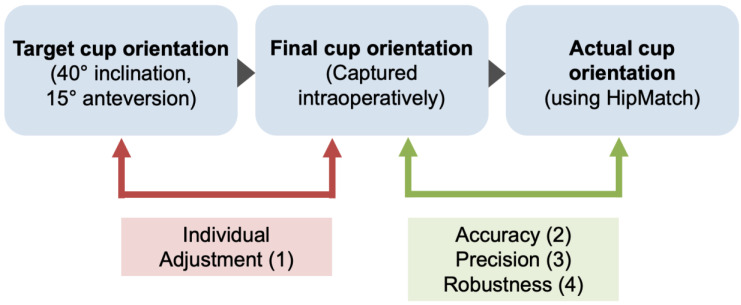
Overview of evaluation of the study results based on the three study questions.

**Figure 2 medicina-58-00832-f002:**
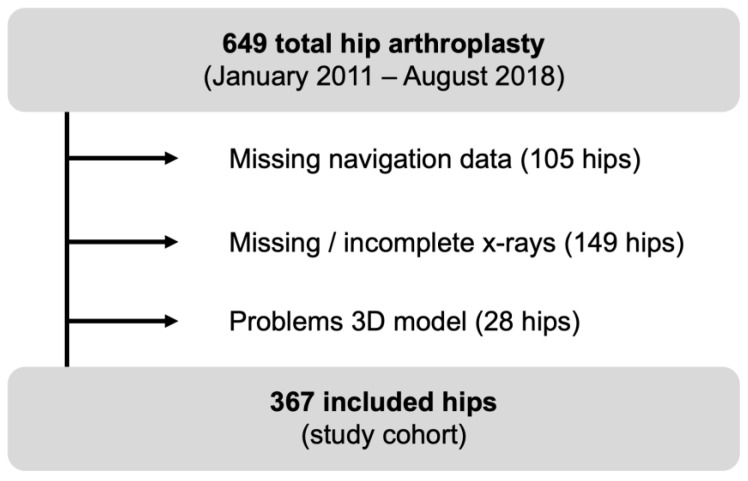
Flow chart of included hips.

**Figure 3 medicina-58-00832-f003:**
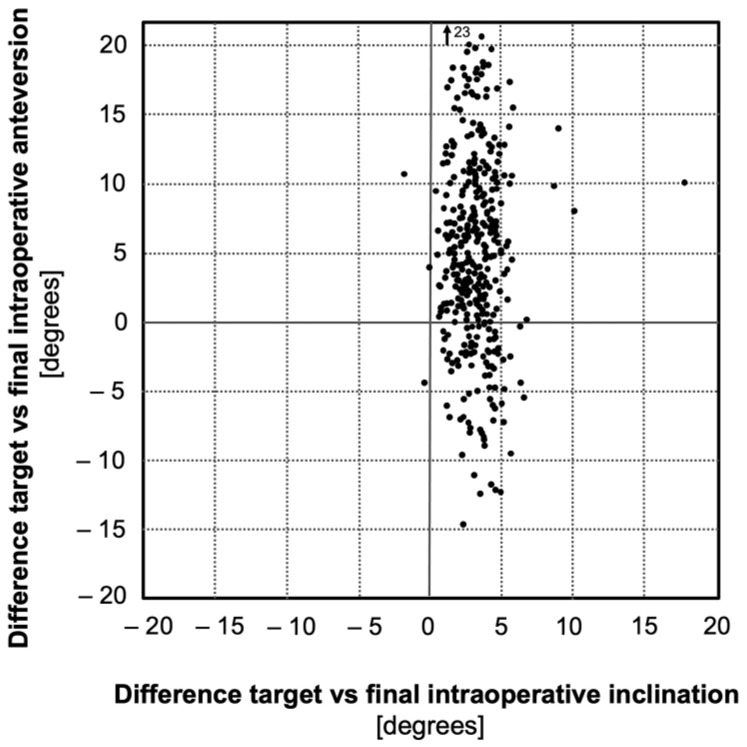
Difference between target and intraoperative positioning assessed by the navigation system (‘individual adjustment’).

**Figure 4 medicina-58-00832-f004:**
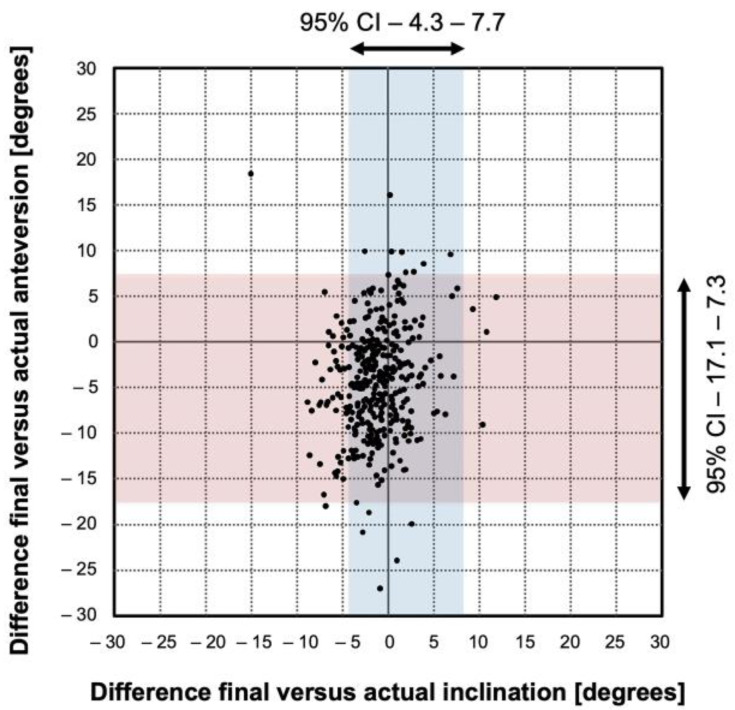
Accuracy and precision of image-free navigated cup positioning. Each point represents one cup. The difference between final and actual inclination and anteversion given as dots in the diagram.

**Figure 5 medicina-58-00832-f005:**
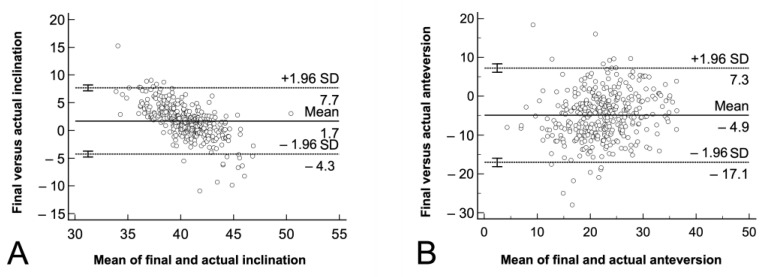
**A**–**B**. The Bland-Altman analysis for the accuracy of image-free navigated cup orientation is shown for (**A**) inclination and (**B**) anteversion. The graphical interpretation shows that the measurement pairs were spread evenly and randomly for both angles. The diagram plots the difference between the final intraoperative and the postoperatively determined actual cup orientation against their averages. The mean accuracy (straight line) and the 95% confidence intervals (dotted line) are shown.

**Table 1 medicina-58-00832-t001:** Demographic and surgery related data.

Parameter	Value(s)
Number of patients	340
Number of hips	367
Age at operation (years)	68 ± 10 (42–91)
Gender (percentage male of all hips)	52
Weight (kg)	78 ± 17 (46–140)
Height (m)	1.70 ± 0.10 (1.50–2.00)
BMI (m/kg^2^)	27 ± 5 (16–48)
Side (percentage right of all hips)	52

Values are expressed as mean ± standard deviation with range in parentheses; THA = total hip arthroplasty; BMI = body mass index.

**Table 2 medicina-58-00832-t002:** Results of cup positioning: Intraoperativ values were assessed with PiGalileo registration system. Calculated 3D values were calculated with HipMatch software based on standard supine pelvic radiographs [3].

Parameter	Degrees
Target inclination (degree)	40
Target anteversion	15
Final intraoperative cup inclination (degree)	41.4 ± 2 (36–52)
Final intraoperative cup anteversion (degree)	20.0 ± 7 (0–38)
Actual cup inclination (degree)	39.7 ± 4 (26–50)
Actual cup anteversion (degree)	24.9 ± 6 (0–41)

Values are expressed as mean ± standard deviation with range in parentheses.

**Table 3 medicina-58-00832-t003:** Summary of literature.

Study	Year	Number of Hips	Type	Navigation System	Postoperative Evaluation	Reference Value Inclination	Range Inclination	Accuracy (*p*-Value)	Reference Value Anteversion	Range Anteversion	Accuracy (*p*-Value)
Tetsunaga et al. [22]	2020	35	CT-based	BrainLAB VectorVision 3.5.2	CT-Based		2.7° ± 2.0°	0.16		2.8° ± 2.6°	<0.001
		35	Accelerometer-based	accelerometer-based navigation	CT-Based		3.3° ± 2.4°			3.4° ± 2.2°	
Okamoto M et al. [23]	2019	113	Portable navigation	Protable navigation	CT-Based	40°	3.1 ± 2.2	0.304	15–20°	2.8 ± 2.3	0.005
		102	Alignment guide	HipAlign	CT-Based	40°	2.9 ± 2.3		15–20°	3.7 ± 2.7	
Yamada et al. [24]	2017	40	CT-based	BrainLAB VectorVision 3.5.2 (2D-3D group)	CT-Based		2.5° ± 2.2°	0.0016		2.3° ± 1.7°	0.0009
		40	CT-based	BrainLAB VectorVision 3.5.2 (PPM group)	CT-Based		4.6° ± 3.3°			4.4° ± 3.3°	
Kalteis et al. [14]	2006	30	Free hand	-	CT-Based	45°			15°		
		30	CT-based	BrainLAB VectorVision 3.0	CT-Based	45°	3.0° ± 2.6		15°	3.3° ± 2.3°	
		30	Imageless	BrainLAB VectorVision 3.0	CT-Based	45°	2.9° ± 2.2°		15°	4.2° ± 3.3°	
Iwana et al. [19]	2013	87	CT-based	Stryker CT-Hip System (older version)	CT-Based	40°	1.8° ± 1.6°	0.98	10°	1.2° ± 1.1°	0.39
Nakahara et al. [25]	2017	49	CT-based	Stryker CT-Hip System (older version)	CT-Based		1.9° ± 1.6°			1.6° ± 1.4°	
		49	CT-based	Stryker CT-Hip System (newer version)	CT-Based		1.2° ± 1.3			1.0 ± 0.8°	
Ybinger et al. [18]	2007	37	Imageless	Plus Orthopaedics PiGalileo	CT-Based		3.5° ± 4.4°			6.5° ± 7.3°	
Parratte et al. [26]	2007	30	Imageless	Hiplogics Universal Protocol (BMI < 27)	CT-Based		4.0° ± 2.8°			3.4° ± 3.6°	
		30		Hiplogics Universal Protocol (BMI ≥ 27)	CT-Based		3.3° ± 3.1			11.6° ± 6.1°	
Ryan et al. [27]	2010	26	Imageless	Ci System	CT-Based		1.8° ± 1.2°			2.0° ± 2.0	
Jenny et al. [17]	2007	48	Imageless	B. Braun Aesculap OrthoPilot	CT-Based		−2° ± 4°			−4° ± 8°	
Fukunishi et al. [15]	2015	83	Imageless	B. Braun Aesculap OrthoPilot	CT-Based	35–45°	3.0° ± 2.6°		15–25°	5.0° ± 3.5°	
Takeda et al. [16]	2017	108	Imageless	B. Braun Aesculap OrthoPilot	CT-Based		3.7° ± 2.7°			6.8° ± 3.6°	
Lin et al. [28]	2011	25	Imageless	Stryker Imageless Navigation System	CT-Based		0.0° ± 2.8°			3.4° ± 3.6°	
Sendtner et al. [20]	2010	32	Imageless	BrainLAB Hip unlimited 5.0	CT-Based		0.4° ± 3.3°			−5.6° ± 6.5°	
Sendtner et al. [20]	2010	32	CT-based		CT-Based						
Hananouchi et al. [29]	2009	40	CT-based	Stryker CT-Hip System (older version) mini posterior approach	CT-Based		2.4° ± 2.0°			2.0° ± 1.4°	
				Stryker CT-Hip System (older version) mini anterior approach	CT-Based		2.0° ± 1.4°			2.7° ± 1.9°	
Hirasawa et al. [30]	2010	56	CT-based	Stryker CT-Hip System (older version)	CT-Based		3.2° ± 2.7°			3.8° ± 3.4°	
Kitada et al. [31]	2011	54	CT-based	Stryker CT-Hip System (older version)	CT-Based		0.4° ± 2.5°			−0.8° ± 4.1°	
Kajino et al. [32]	2012	25	CT-based	Stryker CT-Hip System (older version) for severe pelvic deformities	CT-Based		1.5° ± 1.2°			2.5° ± 1.7°	
		25	CT-based	Stryker CT-Hip System (older version) low grad subluxation	CT-Based		1.4° ± 1.1°			2.7° ± 1.4°	

## Data Availability

The data presented in this study are available on request from the corresponding author. The data are not publicly available due to privacy of patient data.

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
