# Peer review of "Image-Less THA Cup Navigation in Clinical Routine Setup: Individual Adjustments, Accuracy, Precision, and Robustness"

_medicina, 2022, doi:10.3390/medicina58060832_

Round 1

Reviewer 1 Report

Dear authors,

please find my comments regarding your paper. I found the paper well written.

General comments:

There are publications already available correlating intraoperative positionning of navigated hips with postoperative CT's. Therefore, it would have been more accurate and reproducible with higher impact on the conclusions made if the cohort (or at least a subgroup) of intraoperatively navigated hips and consecutively the postop results drawn by the HipMatch software had been finally verified by CT.

This should be discussed more precisely in my opinion and is one further limitation within the methods of the study. A correlation of the results between the used navigation and the (HipMatch) software is made. A certain kind of verification or validation of the results of the used navigation system and/or the HipMatch software can only be drawn by existing literature and not directly by additional postop CT evaluation. If so, please correlate the existing literature of the used navigation system with the results of the study.

Particularly:

Please shorten the paper as informations are given in tables and figures and additionally in the text  (e.g. Figure 2, Table 1). 

Page 3, last chapter: Fifteen hips had previous surgery:..., four surgical hip(s) dislocations... better: open reduction of dislocated THA.

Kind regards

Author Response

Thank you for our comments! Please see the attachment for our author's response. 

Kind regards.

Reviewer 2 Report

I commend the authors for their research entitled "Image-less THA cup navigation in clinical routine setup: Individual adjustments, accuracy, precision and robustness". In their study the authors evaluated image-free navigation (with passive digital reference) with a software program to assess cup orientation based on an anterior-posterior pelvic X-ray in a single-centre single-surgeon case series. The drop-out was 43% (649/282) and there was no standard postoperative CT evaluation of the actual cup position. Saying that the possibilities for (unintentional) bias is significant and should be addressed thoroughly in the Discussion section. Besides, I would suggest the following: (1) define/explain the Tukey method; (2) state how may cups were placed in the "safe zone"; (3) state the follow-up time of the patients included in the study; (4) critically rewrite the Conclusion section (which is the weakest point of the manuscript) - what would you recommend to the users of the image-less navigation systems?   

Author Response

Thank you for your comments. Please see the attachment with the author's responses. 

Kind regards.

Round 2

Reviewer 2 Report

The authors made the necessary changes.